# Hybrid Reinforcement Learning for Optimizing Pump Sustainability in Real-World Water Distribution Networks

## Abstract

This article addresses the pump-scheduling optimization problem to enhance real-time control of real-world water distribution networks (WDNs). Our primary objectives are to adhere to physical operational constraints while reducing energy consumption and operational costs. Traditional optimization techniques, such as evolution-based and genetic algorithms, often fall short due to their lack of convergence guarantees. Conversely, reinforcement learning (RL) stands out for its adaptability to uncertainties and reduced inference time, enabling real-time responsiveness. However, the effective implementation of RL is contingent on building accurate simulation models for WDNs, and prior applications have been limited by errors in simulation training data. These errors can potentially cause the RL agent to learn misleading patterns and actions and recommend suboptimal operational strategies. To overcome these challenges, we present an improved "hybrid RL" methodology. This method integrates the benefits of RL while anchoring it in historical data, which serves as a baseline to incrementally introduce optimal control recommendations. By leveraging operational data as a foundation for the agent's actions, we enhance the explainability of the agent's actions, foster more robust recommendations, and minimize error. Our findings demonstrate that the hybrid RL agent can significantly improve sustainability, operational efficiency, and dynamically adapt to emerging scenarios in real-world WDNs.

## 1 Introduction

Water is essential for sustaining life and supporting various economic activities, making its management a pressing global challenge. Effective water management crucially depends on optimizing pump operations within water distribution networks (WDNs). These networks facilitate the seamless transportation of water from its sources to consumers and serve as the backbone of urban and rural infrastructure (Abkenar et al., 2015).

In the water utility sector, electricity consumption costs have been an ongoing concern for water providers. However, a recent global surge in electricity prices has pushed these costs to the forefront of challenges in operating WDNs (Mala-Jetmarova et al., 2017). Consequently, the effective management of WDN resources not only encompasses safeguarding water availability but also in optimizing its efficient utilization and transportation.

The complexity of this challenge becomes evident when one considers the multifaceted nature of water networks. Demand patterns exhibit fluctuations, energy costs vary, and network conditions evolve in real-time. Traditional rule-based control strategies, which have historically governed pump operations in WDNs, struggle to adapt to the dynamic and uncertain nature of real-world systems. The simultaneous pursuit of conflicting objectives, maximizing energy efficiency while guaranteeing a consistent water supply, adds layers of intricacy.

In response to these challenges, reinforcement learning (RL), a branch of machine learning equipping agents to make sequential decisions through interactions with their environment, emerges as a promising solution (Dong et al., 2020). The application of RL techniques to WDNs unveils a realm of exciting possibilities. It empowers water utilities to dynamically optimize pump opera-

tions, promising significant enhancements in efficiency, reduced energy consumption, and overall system performance.

As this research unfolds, the transformative potential of this work on water management practices becomes evident. It reveals formidable challenges that must be surmounted to realize this vision. Addressing these challenges involves the meticulous design of reward functions, the development of effective exploration strategies, and the utilization of advanced neural network architectures. The overarching objective is to strike a delicate balance between energy efficiency and water supply reliability, guided by the principles of data-driven decision-making.

In summary, this paper delves into the critical intersection of water management, electricity costs, and the application of RL techniques to optimize WDNs. It explores the promise of enhanced efficiency and sustainability while acknowledging the challenges that lie ahead in achieving this vision.

## 2 BACKGROUND

### 2.1 GENETIC ALGORITHMS FOR WDN OPTIMIZATION

In recent years, the field of operating WDNs has witnessed a surge in the application of various optimization methods. These approaches can be broadly categorized into three primary groups: Deterministic methods, Stochastic methods (also known as Metaheuristics), and Hybrid methods, as detailed in Awe et al. (2019).

One of the most popular stochastic optimization methods is genetic algorithm (GA). Several research studies have successfully demonstrated the use of GA for optimizing pump operations in WDNs, resulting in energy cost savings (Boulos et al., 2001; Gupta et al., 1999). However, it's worth noting that standalone use of GA can face challenges. Van Zyl et al. (2004) showed a decrease in convergence speed was observed when using GA exclusively. To address this issue, they proposed a hybrid model that combines GA with a hill-climber search strategy, aiming to overcome the slowdown in convergence. Similarly, Batista do Egito et al. (2023) focused on an optimization model utilizing GA which aims to merge the efficient utilization of reservoirs with the identification of the most effective operational rules for activating pumping systems. Parvaze et al. (2023) examines several advancements in optimizing WDNs. It highlights the utilization of GA as a powerful search method for addressing non-linear optimization challenges. In another study, Sangroula et al. (2022) introduced the "Smart Optimization Program for Water Distribution Networks" (SOP-WDN), which utilizes GA in conjunction with the EPANET hydraulic simulation solver to optimize WDN design.

While utilizing GA for real-world problem-solving, several challenges emerge. These hurdles find effective solutions through the application of RL techniques. GAs involve computationally intensive processes (Chugh et al., 2019; Katoch et al., 2021). RL addresses this challenge by employing learning algorithms that optimize decision-making through interactions with the environment, resulting in reduced computational costs. While GAs may not consistently converge to optimal solutions and offer no guarantee of reaching the best outcome (Katoch et al., 2021), RL agents continually learn from feedback, refining their policies and converging to near-optimal solutions, providing a more dependable approach. Moreover, GAs typically demand a substantial number of samples to yield desirable results, and their time-consuming nature often takes hours to produce real-time pump scheduling results (Hu et al., 2023). In contrast, RL excels in real-time recommendation systems, making rapid decisions based on learned policies, outperforming GAs in this regard.

### 2.2 REINFORCEMENT LEARNING FOR WDN OPTIMIZATION

In recent years, there has been a growing trend in favor of machine learning techniques, particularly RL, which are increasingly recognized as reliable alternatives to traditional optimization methods. For instance, Hu et al. (2023) introduced a deep reinforcement learning (DRL) framework to address real-time pump scheduling challenges within water distribution systems, effectively reducing energy costs without compromising water levels. Similarly, Hajgató et al. (2020) developed an agent employing a dueling deep Q-network, trained to regulate pump speeds based on real-time nodal pressure data. Additionally, Xu et al. (2021) proposed the incorporation of Knowledge-Assisted learning into the framework (KA-RL) to enhance state value evaluation and guide reward function

design. This approach leverages historical data from WDNs to generate optimal trajectories while considering parametric variations.

Although all of the aforementioned methods focus on enhancing performance using RL, this paper goes a step further by introducing a novel hybrid RL approach. This strategy gradually integrates RL-based solutions with existing query-based ones, ensuring long-term reliability and trust. It simultaneously enhances RL-based solutions and maintains system performance. By addressing the challenges associated with GAs and harnessing the power of RL, this hybrid RL approach aims to revolutionize the optimization of WDNs.

## 3 PROBLEM FORMULATION

We define the problem of optimizing pumps in a WDN as a sequential Markov decision process (MDP), which consists of the following key elements:

- For the state space, we investigate two types of agents. For Agent 1, whose objective is to monitor constraint violations, each state $s \in S_1$ is defined by the levels of six unique tanks, resulting in a state dimension of six. For Agent 2, whose objective is to monitor constraint violations and to optimize for cost savings, each state $s \in S_2$ is defined by the levels of six unique tanks, a scalar representing temporal information, and 96 distinct energy tariffs, resulting in a state dimension of 103.

- Each action $a \in A$ is a 6-dimensional vector of real-valued control variables, representing the operational settings of components like pumps and valves. In our final configuration, we utilize six distinct control stations, with each dimension of the action vector corresponding to one of these stations.

- The reward $r_t$ is designed for specific objectives, such as minimizing tank level constraint violations or with an aim to reduce energy costs. There's a preference for simpler step-based rewards for minimizing tank level violations and a balanced multi-objective reward for integrating constraints with energy costs. A detailed reward function will be described in section 4.3.

- The transition function is deterministic, where $s_{t+1} = f(s_t, a_t)$.

- Each episode is terminated at 96 timesteps, equivalent to a 1-day horizon with a 15-minute resolution.

We train an RL agent to learn an optimal policy $\pi_\theta$ in order to maximize the expected discounted rewards:

$$\max_{\pi_\theta} \mathbb{E}_{\tau \sim \pi_\theta} \left[ \sum_{t=0}^{T} \gamma^t r(s_t, a_t) \right], \tag{1}$$

where trajectory $\tau = (s_0, a_0, s_1, a_1, ..., s_T, a_T)$, $\theta$ is the parameterization of policy $\pi$, and $\gamma$ is the discounted factor.

## 4 EXPERIMENTS

### 4.1 DATASET

Data from a water utility company was used to train our RL agent and conduct experiments and assessments. The environment used to train our RL agent is shown in Figure 1. The experiments used this WDN's historical operational data, collected at 15-minute intervals from 2019 to 2023. Key parameters, including tank levels, pump flows/speed, pump power, water demands, and tariff information were available for analysis. The primary objective was to generate optimized control recommendations for components like pumps and valves for the following 24 hours.

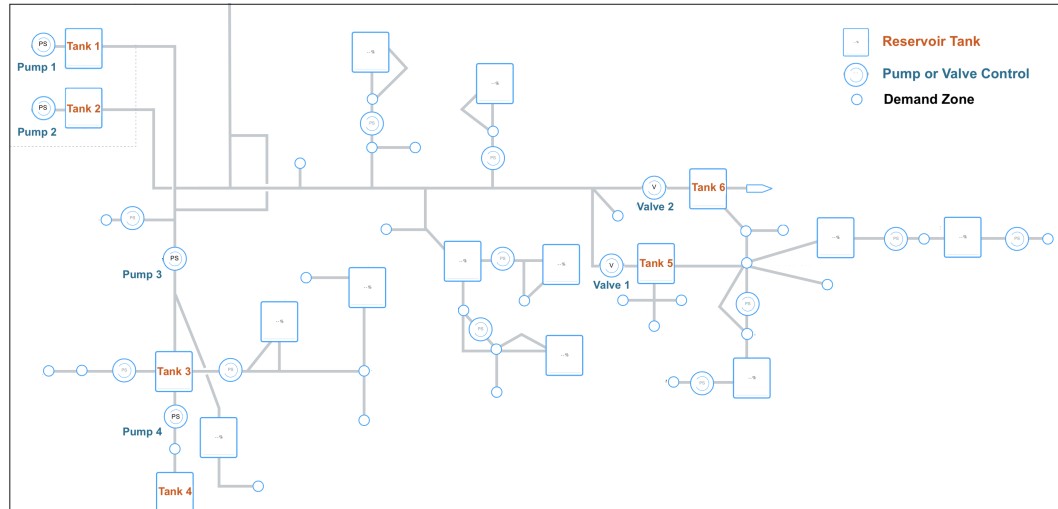

Figure 1: WDN of a water utility company with six pump/valve stations, six reservoirs, and 18 demand zones. A machine-learning based simulation was built for this complex structure to forecast the demand for 24 hours and simulate the resulting tank levels.

## 4.2 RL TRAINING DETAILS

All of our policies are trained using Proximal Policy Optimization (PPO) (Schulman et al., 2017). We use RLlib's[1] implementation of PPO and its default hyper-parameters except `train_batch_size`, with variations $[192, 256, 512, 1024]$ and the configuration with the highest reward was selected. To optimize training efficiency and parallelization, an AWS EC2 instance, namely the g4dn.16xlarge, was employed. This instance accommodated the simultaneous training of multiple agents through the utilization of 15 worker processes, streamlining the experimentation process.

In subsequent sections, we detail the reward function design for optimizing real-time pump scheduling in the WDN using RL.

## 4.3 REWARD FUNCTION:

Reward functions were tailored for specific objectives within the problem domain. Consideration was given to both a singular objective of minimizing tank level constraint violations and a dual objective of minimizing tank level constraint violations while reducing energy costs. Various reward function designs, including step, linear distance-based, and exponential rewards, were explored. The step reward design was ultimately selected for its simplicity and superior performance for the singular objective of minimizing tank level constraint violations.

For the dual objective of reducing tank level constraint violations and minimizing energy costs, a multi-objective reward function was devised according to Algorithm 1. This function integrated the step reward for constraint violation reduction with an energy cost-based reward component. The weights of these reward components were carefully adjusted to achieve a balance between the dual objectives while keeping higher priority for the constraint violation reduction. This approach allowed for the creation of an effective reward function tailored to the research problem, enabling improved decision-making by the agent.

After training our RL agent, we prepared a list of experiments so that we can evaluate its performance in energy saving and constraint violation, especially when introducing various real-world initial conditions and abnormal demand patterns.

The following results are generated using a PPO agent from RLlib with a custom Gym environment.

---

[1]RLlib: https://docs.ray.io/en/latest/rllib/index.html

---

**Algorithm 1** Reward Function for Dual Objectives Agent 2

---

 1: **for** $t \in \{1, \ldots, 96\}$ **do**
 2:     Agent takes an action $a_t$ for state $s_t$
 3:
 4:     $reward\_multiplier \leftarrow 1$
 5:     $constraint\_w, energy\_w \leftarrow 0.7, 0.3$                 ▷ Assign weights for both objectives
 6:     $reward_{\text{constraint}} \leftarrow 0$
 7:     $reward_{\text{energy}} \leftarrow 0$
 8:     $reward_{\text{max}} \leftarrow length(level\_channels) \times (reward\_multiplier)$
 9:     $reward_{\text{min}} \leftarrow length(level\_channels) \times (-reward\_multiplier)$
10:     $cost\_norm \leftarrow$ normalized tariff from 0 to 1
11:
12:     **for** tank_channel in level_channels **do**
13:         $L_t \leftarrow$ level_of_tank_channel
14:         $lb, ub \leftarrow$ Get lower_bound and upper_bound for channel
15:         **if** $lb \leq L_t \leq ub$ **then**
16:             $reward_{\text{constraint}} \leftarrow reward_{\text{constraint}} + reward\_multiplier$
17:         **else**
18:             $reward_{\text{constraint}} \leftarrow reward_{\text{constraint}} - reward\_multiplier$
19:         **end if**
20:     **end for**
21:     $reward_{\text{constraint}} \leftarrow \frac{reward_{\text{constraint}} - reward_{\text{min}}}{reward_{\text{max}} - reward_{\text{min}}}$         ▷ Normalize constraint reward
22:
23:     **for** energy_channel in energy_channels **do**
24:         $E_t \leftarrow$ energy_of_energy_channel
25:         $E\_norm_t \leftarrow \frac{E_t - E_{\text{min}}}{E_{\text{max}} - E_{\text{min}}}$
26:         $cost\_norm_t \leftarrow E\_norm_t \times cost\_norm_t$       ▷ Calculate normalized energy cost
27:         $reward_{\text{energy}} \leftarrow reward_{\text{energy}} + cost\_norm_t$        ▷ Update energy reward
28:     **end for**
29:
30:     $reward(r_t) \leftarrow (constraint\_w \times reward_{\text{constraint}}) + (energy\_w \times reward_{\text{energy}})$
31: **end for**

---

## 5 RL AGENT RESULTS

Figure 2 illustrates reward curves corresponding to distinct agents each with tailored objectives.

### 5.1 AGENTS' PERFORMANCE COMPARISON WITH MEASURED DATA

Table 1 summarizes the key findings of two RL agent configurations: prioritizing constraint adherence (Agent 1) and addressing both constraint compliance and cost optimization (Agent 2). To conduct testing, a total of 330 random sample cases were selected from history ensuring diversity of tank level starting conditions, various demand patterns, and efficiency of pump operations.

Our results demonstrate remarkable improvements in constraint adherence. RL Agent 1, designed for a single objective, achieved a remarkable 90% reduction in total out-of-boundary area compared to historical operational data, along with a notable 93% reduction in number of violations. Equally remarkable was the performance of the dual-objective RL Agent 2, achieving substantial reductions of 88% in total out-of-boundary area and 87% in the number of violations. Despite the additional objective of cost optimization, Agent 2 achieved a 0.2% cost savings, in contrast to a 1.1% loss when cost was not considered. This underscores the high importance of minimizing constraint violations, as reflected in the reward function's design, which places a strong emphasis on this objective.

These findings underscore the tremendous potential of RL agents in addressing multifaceted optimization challenges. By effectively balancing constraints and cost considerations, RL methods offer a powerful solution for industries where system performance and cost-efficiency are paramount. As the demand for adaptable and efficient systems continues to grow, our research highlights the im-

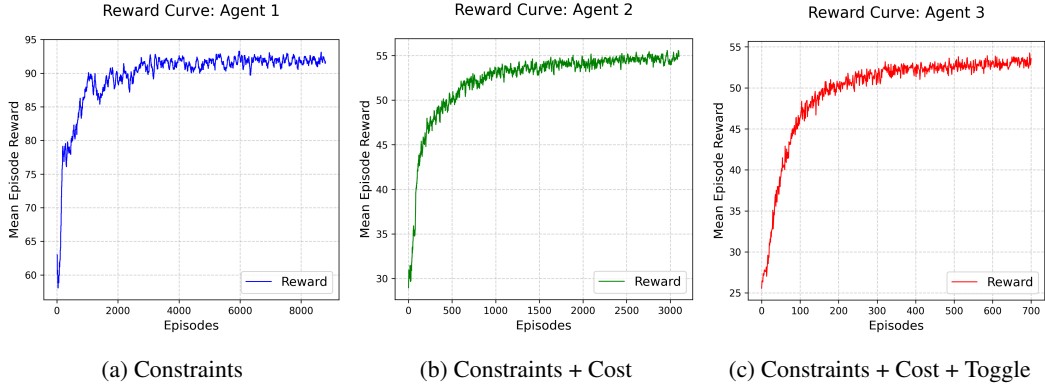

(a) Constraints             (b) Constraints + Cost         (c) Constraints + Cost + Toggle

Figure 2: Reward curves for different Agents serving different objectives. Agent 1 is primarily focused on constraint violations, while Agent 2 is designed with a dual objective, aiming to reduce both constraint violations and overall cost. Agent 3 adopts a dual objective approach with a focus on frame-skipping for efficient toggle management. All these curves demonstrate robust and consistent convergence patterns for all the agents, highlighting the stability of their performance.

Table 1: Performance comparison of two agents with different objectives

| Source of Control Set-Points | Area Out of Boundary | Number of Violation | Cost |
|---|---|---|---|
| Historical Operational Data | 51475 | 9493 | - |
| Agent 1 (constraints) | 5314 (90% improvement) | 661 (93% improvement) | 1.1% loss |
| Agent 2 (constraints + cost) | 5974 (88% improvement) | 1192 (87% improvement) | 0.2% savings |

pact of RL agents in optimizing complex systems, ensuring compliance with critical constraints, and minimizing operational costs.

## 5.2 FRAME-SKIPPING TO HANDLE TOGGLE-COUNT REQUIREMENT

The agent's prior approach involved issuing action recommendations every 15 minutes for the upcoming 24-hour period, which caused an undesirable lack of control smoothness. This frequent and substantial alteration in pump operations and control introduced the potential for unforeseen system issues. In response to this challenge, we explored the concept of "frame-skipping" (Kalyanakrishnan et al., 2021). This strategic methodology imposes restrictions on the agent's action changes, governed by a predefined "toggle-count," thereby ensuring a more consistent set of control recommendations. Frame-skipping deliberately omits action decisions at specified intervals, allowing the agent to maintain its existing chosen action within the n-frame window. Moreover, the environment updates the system's state only after a specific number of frames and provides rewards within a corresponding frame window, enhancing the agent's proficiency in making accurate decisions for the subsequent frame window.

The results of an additional 40 test cases are presented in Table 2, illustrating the impact of setting the frame-skipping window to two hours and permitting the agent to issue a maximum of 12 actions over a 24-hour time frame. This marks a noteworthy reduction from the 96 actions allowed in the absence of toggle-count constraints. While this additional constraint does affect performance with regard to constraint violation objectives, it successfully achieves the objective of ensuring smoother control. This adaptive approach significantly enhances system stability and efficiency, effectively reducing the risk of unexpected disruptions in pump and valve operations, all while optimizing the performance of real-time WDNs.

Table 2: Performance comparison of the agent without toggle-count vs. with toggle-count

| Source of Control Set-Points | Area Out of Boundary | Number of Violation | Cost Savings |
|---|---|---|---|
| Historical Operational Data | 6974 | 1185 | - |
| Agent 2 (without toggle count) | 1136 (84% improvement) | 239 (80% improvement) | 0.56% |
| Agent 3 (with toggle count) | 1958 (72% improvement) | 471 (60% improvement) | 0.25% |

## 6 HYBRID RL

Leveraging RL to control WDNs necessitates an emphasis on the interpretability and stability of recommendations, particularly given the inherent risks of disruptions or violations of physical constraints. To engender trust and promote the seamless integration of RL-derived recommendations, we have embraced a hybrid approach. This methodology seeks to gradually incorporate RL into pump optimization strategies, grounded in a historical baseline given by a query model. In this combined framework, the query-based model acts as a foundation for pump optimization rooted in historical data and established rules. Concurrently, hybrid RL formulates control recommendations from the RL agent for a subset of the upcoming 24-hour window based on distinct criteria and subsequently melds these with the outputs from the query-based model. This synergy between machine learning and empirical data allows us to capitalize on the merits of RL, while ensuring that recommendations not only adhere to the physical constraints of the system but also remain within the comfort zone of historically routine operations.

To fully understand the advantage of using RL alongside results from a query-based model for building trust, it is essential to describe the workings of the query model itself. The recommendations provided by this model are grounded in historical data that closely matches the current state of the system. Factors such as current tank levels, network volume, and forecasted demand are observed and matched against historical operational data. The resulting control points tied to the closest matching historical state are then presented as the primary recommendations for the simulation. However, even with this control-point input, the simulator's outcome might still breach certain rules and constraints due to cumulative simulation error, as shown in Figure 3a, and the possibility that the most proximal historical matches to the current state were not similar enough.

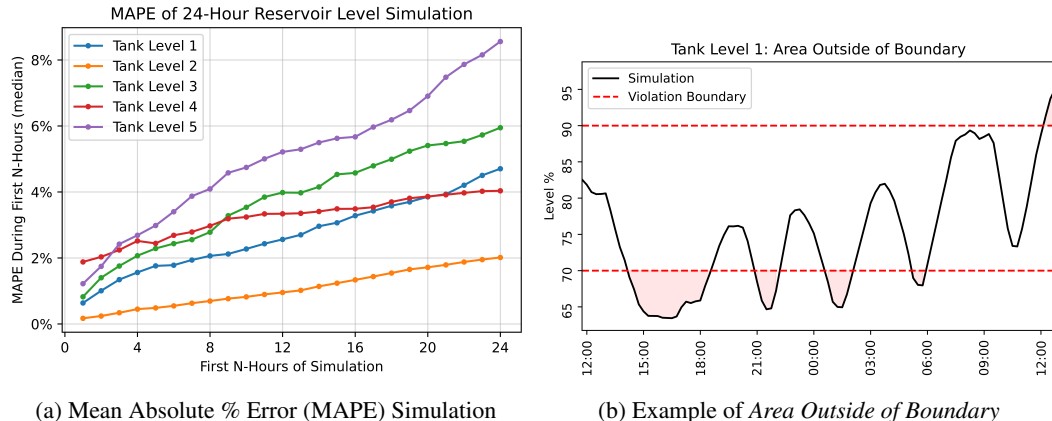

(a) Mean Absolute % Error (MAPE) Simulation    (b) Example of *Area Outside of Boundary*

Figure 3: Metrics used in evaluating performance of simulation and hybrid RL experiments. MAPE of simulation shows error range comparing simulated tank levels vs. historical operational data when simulating a subsequent 24-hour period. *Area Outside of Boundary* serves as a metric for success in adhering to operational constraints. Reducing this metric quantifies progress and allows comparison between hybrid RL strategies.

When the historical recommendation generates a simulation that violates physical constraints, the RL agent can be utilized to inject its recommendation and align the recommendation with learned boundaries. The inherent question then becomes the optimal timing, location, and extent of RL agent injection with the simulation result from historical matching. We conducted several experiments to determine the best answers to these questions, resulting in a hybrid RL recommendation effectively leveraging both paradigms.

A dataset of time series that included constraint violations from simulation was collected and the RL agent setup to inject its recommendation. The primary metric employed for evaluating the outcomes of the experiments was the minimization of the *area outside of boundary*, specifically, the aggregate distance from values exceeding the boundaries to their closest permissible boundary, shown in Figure 3b. By monitoring this metric during and after injection, we could analyze different strategies to both improve the simulation and give stable recommendations for future behavior. Multiple scenarios were tested to observe the hybrid RL agent behavior and evaluate whether the agent was reducing overall constraint violation results and improving the metric:

**1. Untargeted Start/End Time:** Constraint violations within a sample of 24-hour periods were analyzed. The RL agent provided recommendations at static points, specifically not targeting the start and end times of these violations. We examined two separate injection intervals: 0-2 hours and 12-14 hours. The aim was to determine whether total overall violations in the 24-hour span were improved. This approach offered insights into understanding the RL agent's performance under a straightforward strategy, and the results demonstrated improvement even in this untargeted setting. Notably, Figure 4 shows the injections during the hours 12-14 demonstrated superior performance (-16.5% during and -13.1% after injection) compared to those between hours 0-2 (-12.5% during and -9.4% after injection). This can be attributed to the higher likelihood of the untargeted injection being more proximal to the violation itself.

**2. Targeted Start/End Time:** In this experiment, the RL agent's recommendations were implemented exclusively during the constraint violation time window. Figure 4 shows violations during the injection period were further reduced from untargeted injection (-16.5% to -22.4%). An insignificant worsening (-13.1% to -12.8% reduction) was noted in the post-injection metric. This strategy's ability to decrease simulation error while not significantly affecting the overall recommendation established it as the foundational approach for subsequent experiments.

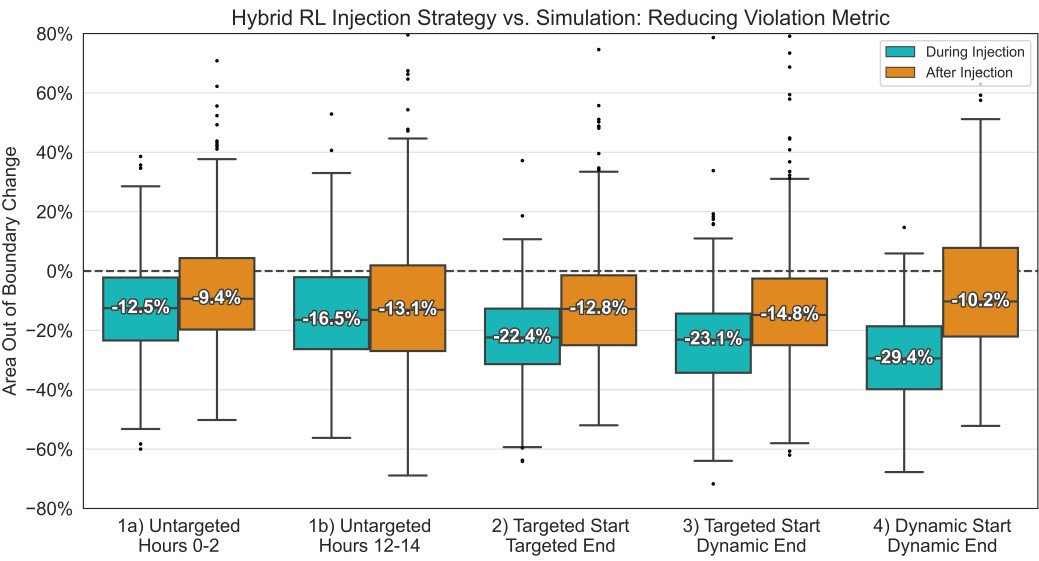

Figure 4: Reducing constraint violations across hybrid RL strategies. Starting with an untargeted strategy (1a, 1b), then targeted (2), combined targeted-dynamic (3), and finally fully dynamic (4). While violations during injection consistently improve, post-injection metrics deteriorate with the fully dynamic approach, underscoring the benefit of only dynamically optimizing the end time.

**3. Targeted Start/Dynamic End Time:** Results from the previous experiment showed that while violation periods were improved by targeted injection, new violations were sometimes introduced in subsequent post-injection time. To rectify this behavior, we introduced a dynamic ending point leveraging knowledge of the simulation's behavior. It was observed that the simulation behavior demonstrated a linear relative shift and not an absolute change when varying the initial point values. This behavior facilitated rapid prediction and analysis of post-injection simulation. Figure 4 shows that an optimal ending time for RL injection was found that reduced during-injection violations further (-22.4% to -23.1%), while also improving post-injection violations (-12.8% to -14.8%).

**4. Dynamic Start/Dynamic End Time:** Building on the observed improvements from introducing a dynamic end time, it was a natural next step to consider adjusting the injection start time dynamically in hopes of further minimizing recommendation violations. While linear adjustments in the simulation proved effective for optimizing the RL injection end time, this method was unsuitable for altering the RL agent's start time. Consequently, any variations of start time before the violation meant a whole new inference by the RL agent. By searching these scenarios to find an optimal injection start time, we observed an improved reduction in during-injection metrics (-23.1% to -29.4%). However, a significant degradation of post-injection metrics emerged (-14.8% to -10.2%), as shown in Figure 4. Introducing the injection prematurely caused destabilization in the relationship between agent and simulation, as measured by post-injection metrics. Subsequent work will explore the injection timing not solely based on the start time but potentially as a function of the boundary value itself or the series trend preceding violation.

## 7 CONCLUSION

The integration of reinforcement learning (RL) into our methodology represents a substantial advancement over conventional techniques to optimize operations in water distribution networks (WDNs). RL's inherent strength lies in its dynamic adaptability to evolving conditions and its capability for real-time decision-making. RL enables the precise determination of optimal control set-points which are responsive to the fluctuations of demand, energy costs, and network adjustments within WDNs. Moreover, the efficiency of RL in terms of sample utilization accelerates its convergence towards optimal solutions, resulting in a significant enhancement in overall operational efficiency and performance in WDNs. By leveraging critical inputs such as initial tank levels, tariff structures, and demand forecasts, our methodology identifies control set-points that strike a balance between cost savings and adherence to operational constraints.

Our comprehensive hybrid RL approach combines RL-based techniques with a query-based warm start strategy, empowering us to deliver real-time optimal control set-points for WDNs that are grounded in historical operational data. The strategic combination of a query-based warm start method with RL harnesses their respective advantages, offering operators dependable and economically viable recommendations for optimizing pump control in WDNs, thus ensuring efficient operation while adhering to operational constraints. During the deployment phase, the gradual introduction of RL-based control set-points not only serves to bolster the robustness of the final control strategy, but integrates with the current operational routine to account for potential errors that may arise during the simulation process. Ultimately, the hybrid RL approach combines the reliability of historical data with the adaptability of reinforcement learning, providing an unparalleled solution for enhancing the efficiency and reliability of WDN operations.

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
