# OpenReview forum: "Hybrid Reinforcement Learning for Optimizing Pump Sustainability in Real-World Water Distribution Networks"
_ICLR.cc/2024/Conference — ICLR 2024 Conference Withdrawn Submission_

### Official Review · Reviewer_uinH · 2023-11-03

**Soundness:** 2 fair
**Presentation:** 2 fair
**Contribution:** 1 poor
**Rating:** 1
**Confidence:** 3

**Summary:**

This paper presents a hybrid reinforcement learning methodology for optimizing pump sustainability in real-world water distribution networks (WDNs). The authors address the challenges of traditional optimization techniques and propose an improved approach that combines the benefits of reinforcement learning with historical data. The findings demonstrate that the hybrid RL agent significantly improves sustainability, operational efficiency, and adaptability in real-world WDNs.

**Strengths:**

None.

**Weaknesses:**

1. The novelty of this paper is poor. It seems like just introducing a practical RL algorithm PPO into WDN problem.
2. This paper may be more suitable for a professional conference in WDN rather than ICLR.
3. Safety-concerned control task could be more suitable with Safe RL algorithms.
4. The writing could be much improved.

**Questions:**

See the weaknesses.

---

### Official Review · Reviewer_WJdj · 2023-11-06

**Soundness:** 2 fair
**Presentation:** 1 poor
**Contribution:** 1 poor
**Rating:** 3
**Confidence:** 2

**Summary:**

The paper presents an innovative approach to optimizing water distribution networks (WDNs) using a hybrid reinforcement learning (RL) methodology that leverages historical data to generate optimal trajectories while considering parametric variations. This method gradually integrates RL-based solutions with existing query-based warm start methods, ensuring long-term reliability and trust.

**Strengths:**

1. Provides a very practical industrial application scenario.

**Weaknesses:**

1. The organization of this paper is unclear, the methodology section (Sec 6 Hybrid RL) is located after the experiments section, which makes readers confused.
2. It seems like pure application work, I am a bit concerned that the idea of inserting a vanilla PPO algorithm into a specific industrial pipeline can not support enough novelty in a machine learning conference.
3. Lack of enough ablation or comparison with other methods.

**Questions:**

I have no further questions.

---

### Official Review · Reviewer_E3Kq · 2023-11-09

**Soundness:** 4 excellent
**Presentation:** 4 excellent
**Contribution:** 3 good
**Rating:** 3
**Confidence:** 5

**Summary:**

This submission uses RL to address the pump-scheduling problem in water distribution networks (WDNs). First, the authors demonstrate the performance of agents trained using PPO on a WDN environment using a particular WDN's historical operational data, and with reward functions designed to optimize constraints and costs; these agents show substantial improvement in ensuring constraint violations, in particular. The authors further test out a hybrid approach that switches between the previously-used baseline model (a query model) and the RL model, with the aim of engendering trust; they evaluate a few different strategies for when the RL model's recommendations should be injected.

**Strengths:**

This paper addresses an important task of real-world significance: The management of water distribution networks. This problem will only become increasingly important as, e.g., climate change affects both water supply and electricity prices (given that water distribution networks rely on electricity to operate).

The use of RL is well-motivated, and empirical results on a simulation environment show substantial performance improvements compared to the currently-used query-based model.

The introduction of a hybrid RL methodology switching between the previously-used model and the new RL model is interesting, as this paradigm could indeed be helpful in the integration of RL into systems where operators may be hesitant to rapidly adopt new technologies.

The paper is well-written and easy to follow.

**Weaknesses:**

While there is nothing significantly wrong with the paper itself, I unfortunately do not think it is a fit for ICLR. In particular, to be presented at ICLR, a paper must present either (a) a strong methodological contribution in ML or (b) a strong applied contribution that provides further insight and directions to researchers in the ICLR community.

In terms of ML methodological contribution: The introduction of the hybrid approach is genuinely interesting, but would need to be further developed and tested out to truly constitute a strong ML methodological contribution. In particular, the different metrics for switching between the query model and RL actions seem somewhat ad hoc, which is potentially very reasonable in the context of the application at hand, but would need to be developed in a more principled/structured way to constitute a strong methodological contribution in ML.

In terms of applied contributions providing insight to the ICLR community: My evaluation here is along the lines of two questions:
* *Does the paper provide useful insight on considerations associated with using ML in real-world settings?* - While the paper does discuss considerations like the need to satisfy constraints, these are not described in sufficient detail. Such detail and accompanying discussion would be necessary to shed light on ways that RL methods could be fundamentally improved to, e.g., better satisfy the kinds of constraints at hand in real-world settings.
* *Is the end-to-end real world implementation convincing from the application perspective - e.g., is it actually deployed on a real-world system, and does it appropriately fulfill application-specific criteria?* - While the paper's evaluation uses real-world data from a WDN, it is not clear to me how realistic the simulation environment is or how close to deployment the methods are.

With all that said, I really do commend the authors' work. My comments above are truly only about fit for this venue, and not a statement on the objective quality of this work (notably, complex ML methods are not always needed!).

**Questions:**

* Can the authors shed additional light on the kinds of operational constraints that agents in this application domain must satisfy?
* Can the authors shed additional light on the realism of the simulation environment, and the maturity/closeness-to-deployment of the RL methods presented?

---

### Official Review · Reviewer_V1rK · 2023-11-10

**Soundness:** 2 fair
**Presentation:** 1 poor
**Contribution:** 1 poor
**Rating:** 3
**Confidence:** 4

**Summary:**

This paper aims to address the control issues of real-world water distribution networks (WDNs). Specifically, the paper introduces reinforcement learning to enhance decision-making speed and performance, with a finely tailored reward function to aid the training of reinforcement learning, demonstrating significant improvements over historical data. The article also explores a control approach that integrates reinforcement learning with historical data, improving the efficiency and reliability of WDN operations.

**Strengths:**

1. The paper applies reinforcement learning to WDN control, showing better performance in avoiding violations and saving energy.

**Weaknesses:**

1. The experiment lacks persuasiveness as it does not compare with state-of-the-art baseline algorithms, making it difficult to demonstrate the high performance and robustness of the proposed method.
2. The description of the experimental section is not clear enough, causing confusion for the reader, especially those parts concerning Figure 4. The authors should revise the experimental description to be clearer.

**Questions:**

1. What is the setting of simulation in the paper? The article does not explain how the simulation is implemented, which is confusing here.